# Is Technological Progress Selective for Multiple Pollutant Emissions?

**DOI:** 10.3390/ijerph18179286

**Published:** 2021-09-02

**Authors:** Weijiang Liu, Mingze Du

**Affiliations:** 1Center for Quantitative Economics, Jilin University, Changchun 130012, China; liuwj@jlu.edu.cn; 2Business School, Jilin University, Changchun 130012, China; 3Northeast Revitalization and Development Research Institute, Jilin University, Changchun 130012, China

**Keywords:** technological progress, technological progress bias, pollutant emissions, inverted U-shape

## Abstract

Current research on technological progress does not focus on whether there is a biased selection of technological progress based on the resulting pollutant emissions and the emission reduction effect. This paper measures green total factor productivity for 30 provinces in China from 2004–2018 and tests whether technological progress is selectively biased towards the pollutants emitted. The results find a selective bias of technological progress on pollutant emissions, and there is also heterogeneity in the selective bias across regions. The current level of technological progress is on the right side of the inverted U-shaped inflection point for SO_2_ and PM_2.5_ and the left side of the inverted U-shaped inflection point for CO_2_. The improvement of technological progress can reduce the emissions of SO_2_ and PM_2.5_. Still, the results indicate that the reduction effect of these two pollutants originates from the treatment process rather than reducing the source of the production side. The inability of technological advancement to reduce CO_2_ emissions suggests some carbon lock-in in China’s technological advancement. The Chinese government should increase the proportion of new energy applications and reduce the production methods of polluting industries to reduce pollutants effectively.

## 1. Introduction

China’s economy has been reformed and opened up for more than 40 years and has made brilliant achievements in the economic field. However, since the reform and opening up, China’s economy has been in a rough economic development model for many years, making China’s environmental pollution problem increasingly serious. According to the World Health Organization (WHO), air pollution causes nearly 7 million deaths worldwide each year. As one of the most polluted countries globally, China has endured haze pollution, acid rain, and excessive greenhouse gas emissions that continue to plague the country in the fall and winter of recent years [1,2]. According to the Cost of Pollution in China 2007, a study completed by the World Bank in collaboration with the Chinese government, severe air pollution in China causes 350,000–400,000 deaths per year due to indoor air pollution [3]. Currently, China is among the world leaders in PM_2.5_, SO_2_, NO_2_, and greenhouse gas emissions. More seriously, in 2011 and 2015, several Chinese provinces experienced PM_2.5_ explosions with pollution levels reaching up to 1155 μg/m^3^, and serious environmental pollution problems will pose a significant threat to the health of residents [4,5]. Data in the 2017 China Ecological Environment Status Bulletin also show that the percentage of cities with excessive environmental pollution in China is as high as 70.7%. The rate of days with PM_2.5_ and PM10 as the primary pollutants was 74.2% and 20.4%, respectively [6]. Numerous pieces of evidence show that the increasing environmental pollution has induced a continuous increase in the rate of various chronic diseases among Chinese residents. The environmental pollution problem poses a significant threat to the quality of China’s economic development [7,8,9].

In the face of severe environmental problems and the need to transform the economic development model, China has continued to transition to a greener economy in recent years. The key is to promote technological innovation to achieve pollution reduction and sustainable economic development [10]. However, the answer to whether the promotion of technological progress can effectively reduce environmental pollution is inconclusive. By studying technological progress and environmental pollution in five developing countries—Brazil, Russia, India, China, and Mexico—scholars found that technological progress leads to increased pollution in developing countries when their GVCs are below a threshold. Otherwise such technological progress can reduce emissions [11]. Some scholars have also analyzed Chinese industrial data and found that technological progress can reduce pollution [12]. Some studies have shown that technological progress from different sources has different effects on pollution, with independent innovation having a significant inhibitory effect on haze pollution, while technology introduction exacerbates haze pollution [13]. Neutral technological advances and labor-saving technological advances are beneficial to haze emission reduction. In contrast, capital-saving technological advances have insignificant effects on haze pollution, and energy-saving technological advances cannot effectively reduce haze pollution due to the energy rebound effect [14].

The invention of the steam engine set off a wave of the industrial revolution. It also brought about quite serious pollution events such as the London toxic fog that caused thousands of deaths [15,16]. Some studies have shown that the economy’s structure based on fossil energy sources has also created certain carbon lock-in and energy rebound effects [17,18,19,20]. Some scholars have argued that the carbon lock-in effect is one of the major obstacles in the transition to a low-carbon economy and that carbon lock-in can lead to greater welfare losses in the absence of regulatory conditions [18]. This was also found when analyzing the energy consumption behavior of urban households [19]. This is because the path of industrial countries when building industrial systems is locked in the fossil energy-based energy system. Thus, technological progress will rely more on that path, and the scale effect of technological progress will make the production efficiency and production scale further expand. The rise in demand for energy leads to more pollution problems, so even if technological progress occurs, it is not always possible to reduce environmental pollution [14,20]. Technological progress pushing enterprises to expand their production scale, leading to more energy consumption, can bring more serious environmental pollution problems. The existing fossil energy-based technology system also forms a certain obstacle to the promotion and application of green technological progress. For example, studies of urban residential electricity consumption in China show that there is indeed an energy rebound effect [21]. An analysis of the rebound effect of energy consumption in China shows that this phenomenon does exist in China, where technological advances have improved energy efficiency, but the rebound effect has made energy savings less effective than expected [22]. Some scholars have shown by estimation that the rebound effect in China during 1981–2011 was between 30% and 40% [23]. However, some scholars’ studies have shown that technological progress can significantly improve environmental quality and reduce environmental pollution. Unlike previous industrial revolutions, technological progress in recent years has been about efficiency improvements. The application of green energy sources such as photovoltaics and wind power has led to technological progress toward cleaner and energy-efficient technologies. China has made great progress in clean energy technology [24], and studies have shown that the development and application of clean energy technology can effectively reduce pollution [25,26]. It has been found that through the analysis of the impact of China’s trade on technological progress, imports from developed countries such as Europe and the United States increase the rate of green technological progress. In contrast, imports from developing countries decrease the rate of green technological progress [27]. Moreover, some scholars have studied the impact of environmental regulation, fiscal decentralization, and foreign investment on green technological progress in order to achieve a reduction in pollution emissions [28,29,30,31,32].

However, in the face of the multiple pollutants embedded in the environmental pollution brought about by the rough economic development, can the promotion of technology effectively reduce the emissions of all pollutants? Is the relationship between the action of air pollutants represented by PM_2.5_, SO_2_, and CO_2_ levels and technological progress consistent with the inverted U-shaped relationship of the environmental Kuznets curve [33,34]? Although there are studies related to the social causes of pollutants such as ozone and PM_2.5_ [35,36], is there a bias of technological progress towards the reduction of different kinds of pollutants? There are few relevant studies on these questions, especially on whether there is a certain bias of technological progress towards the reduction targets of multiple pollutants. Related studies have mostly focused on the relationship between technological progress in input bias and individual pollutants [29,36]. Biased technological progress is currently a hot research topic in the academic community [37,38,39,40]. Most of the literature has focused on the bias of technological progress on input-based technological progress, with the aim of verifying whether the preference of input factors contributes to technological progress, without considering the issue of multiple outputs [41,42,43]. Most of the available studies on output-based technological progress bias have focused on undesirable outputs represented by CO_2_ and have not paid attention to the bias of technological progress for multiple types of undesirable outputs [29,44,45,46].

In this paper, based on the existing studies, we measure the green total factor productivity of Chinese provincial panel data and decompose the green technological progress from it; secondly, we verify the possible inverted U-shaped relationship between multiple air pollutants and technological progress. Then, this paper analyzes the bias of technological progress for emission reduction among different pollutants in detail to investigate whether there is a preference for different types of pollutants and whether there is heterogeneity in the bias of technological progress among different provinces in China. Finally, the selective bias of technological progress on pollution emissions is empirically analyzed by means of an econometric model to verify the analysis made in the previous paper.

Compared with previous studies, the marginal contributions of this paper are as follows. (1) Verifying whether the inverted U-shaped relationship between multiple air pollutants and technological progress holds. (2) Decompose technological progress and analyze the bias of output-based technological progress for different kinds of air pollutants. (3) Analysis of the possible bias of pollutant emission reduction in different provinces in China and attempt to explain the bias that causes different biases among different provinces. (4) To verify the bias of output-based technological progress for different types of air pollutants by using econometric models. By solving the above problems, this paper hopes to provide suggestions for China to adjust its environmental protection policies.

## 2. Methods and Data

This section presents the econometric models involved in this paper, where the DEA-SBM (Slack-Based Measure, SBM) model is used to measure green total factor productivity and describes the process of decomposing it into technological progress and technological progress bias. The regression model is a validation of the inverted U-shaped relationship between multiple pollutants and technological progress and empirical analysis of socioeconomic factors.

### 2.1. The DEA-SBM Model

Among the methods for measuring total factor productivity, the DEA model approach can be used for multiple inputs and outputs and does not require constructing a production function to estimate the parameters, thus avoiding the errors associated with an artificially set production framework. Therefore, the DEA model is selected in this paper to estimate the required total factor productivity. The traditional DEA model fails to solve the problem of slack variables better when measuring efficiency evaluation. Therefore, in this paper, we adopt the SBM model proposed by Tone [47], with undesired outputs, which can consider the relationship between inputs, outputs, and undesirable outputs in an integrated way and can better solve the slack problem in efficiency evaluation.

In this study, we use Chinese province panel data, wherein each of the 30 provinces is made a production decision unit DMU to construct the optimal production frontier of China in each period. Each province uses M kinds of inputs to obtain S kinds of desired outputs and I kinds of non-desired outputs, then the DEA-SBM model can speak about the production process of Chinese provinces as:(1)$$p(x)=\left\{\left(x,y,b\right)\left|:\right.x\geq X\lambda ,y\geq Y\lambda ,b\geq B\lambda ,\lambda \geq 0\right\}$$

According to Equation (1), the DEA-SBM model for undesired output can be written as:(2)$$\begin{array}{l} \min\rho_{k}=\frac{1-\frac{1}{m}\sum_{i=1}^{m}\frac{s_{i}^{x-}}{x_{ik}}}{1+\frac{1}{p+q}\left(\sum_{r=1}^{p}\frac{p_{r}^{y+}}{y_{rk}}+\sum_{t=1}^{q}\frac{z_{t}^{b-}}{b_{tk}}\right)}\\ s.t.X\lambda +s^{x-}=x_{k^{\prime }}Y\lambda -s^{y+}=y_{k^{\prime }}B\lambda +s^{b-}=b_{k^{\prime }},\lambda \geq 0,s^{x-},s^{y+},s^{b-}\geq 0 \end{array}$$
where $s^{x-}$, $s^{y+}$, $s^{b-}$ represent the slack values of the input, good output, and bad output, respectively. $x_{mk}$, $y_{pk}$, $b_{qk}$ represent the *m*th input of the *k*th DMU, the *p*th desired output, and the *q*th undesired output, respectively. $\rho_{k}$ is a variable between 0 and 1 representing the efficiency of the *k*th DMU environment, where less than 1 means that the *k*th DMU is inefficient.

This study constructs the Malmquist index distance function in conjunction with the SBM model dealing with undesirable output. According to the Malmquist exponential decomposition method, the total factor productivity (*TFP*) growth rate is decomposed into technological change (*TC*) and efficiency change (*EC*). We further decompose technological change into output-biased technological progress (*OBTC*) and input-biased technological progress (*IBTC*) and magnitude of technological change (*MATC*).

First, assuming that $\rho_{k}^{t}\left(x^{t+1},y^{t+1},b^{t+1}\right)$ and $\rho_{k}^{t+1}\left(x^{t+1},y^{t+1},b^{t+1}\right)$ are the efficiency of the *k*th DMU in period *t* to *t* + 1, China’s green Malmquist productivity index is defined as follows:(3)$$TFP_{k}^{t,t+1}={\left[\frac{\rho_{k}^{t}\left(x^{t+1},y^{t+1},b^{t+1}\right)}{\rho_{k}^{t}\left(x^{t},y^{t},b^{t}\right)}\times \frac{\rho_{k}^{t+1}\left(x^{t+1},y^{t+1},b^{t+1}\right)}{\rho_{k}^{t+1}\left(x^{t},y^{t},b^{t}\right)}\right]}^{\frac{1}{2}}$$

$TFP_{k}^{t,t+1}$ > 1 indicates that the green *TFP* is growing from period *t* to period *t* + 1, and $TFP_{k}^{t,t+1}$ < 1 indicates that the green *TFP* is reduced from period *t* to period *t* + 1. According to the Malmquist exponential decomposition method by Fare [48], the *TFP* growth rate is decomposed into technological change and efficiency change as follows:(4)$$TFP_{k}^{t,t+1}=TC_{k}^{t,t+1}\times EC_{k}^{t,t+1}={\left[\frac{\rho_{k}^{t}\left(x^{t},y^{t},b^{t}\right)}{\rho_{k}^{t+1}\left(x^{t},y^{t},b^{t}\right)}\times \frac{\rho_{k}^{t}\left(x^{t+1},y^{t+1},b^{t+1}\right)}{\rho_{k}^{t+1}\left(x^{t+1},y^{t+1},b^{t+1}\right)}\right]}^{\frac{1}{2}}\times \frac{\rho_{k}^{t+1}\left(x^{t+1},y^{t+1},b^{t+1}\right)}{\rho_{k}^{t}\left(x^{t},y^{t},b^{t}\right)}$$

$TC_{k}^{t,t+1}$ denotes the shift of the *k*th DMU in the period *t* to *t* + 1 of the technological change, i.e., the technological frontier. $EC_{k}^{t,t+1}$ indicates a change in relative efficiency.

After decomposed *TFP* Fare [49], decomposing *TC* into an index of the magnitude of technological change and an index of technological bias (*BTC*), the technology bias index can be decomposed into input-biased technological progress and output-biased technological progress indices as follows:(5)$$\begin{array}{cc} \; & TC_{k}^{t,t+1}={\left[\frac{\rho_{k}^{t}\left(x^{t},y^{t},b^{t}\right)}{\rho_{k}^{t+1}\left(x^{t},y^{t},b^{t}\right)}\times \frac{\rho_{k}^{t}\left(x^{t+1},y^{t+1},b^{t+1}\right)}{\rho_{k}^{t+1}\left(x^{t+1},y^{t+1},b^{t+1}\right)}\right]}^{\frac{1}{2}}\\ \; & =\frac{\rho_{k}^{t}\left(x^{t+1},y^{t+1},b^{t+1}\right)}{\rho_{k}^{t+1}\left(x^{t+1},y^{t+1},b^{t+1}\right)}\times {\left[\frac{\rho_{k}^{t}\left(x^{t},y^{t},b^{t}\right)}{\rho_{k}^{t+1}\left(x^{t},y^{t},b^{t}\right)}\times \frac{\rho_{k}^{t+1}\left(x^{t+1},y^{t+1},b^{t+1}\right)}{\rho_{k}^{t}\left(x^{t+1},y^{t+1},b^{t+1}\right)}\right]}^{\frac{1}{2}}\\ \; & =MATC_{k}^{t,t+1}\times BTC_{k}^{t,t+1} \end{array}$$
(6)$$\begin{array}{cc} \; & BTC_{k}^{t,t+1}={\left[\frac{\rho_{k}^{t}\left(x^{t},y^{t},b^{t}\right)}{\rho_{k}^{t+1}\left(x^{t},y^{t},b^{t}\right)}\times \frac{\rho_{k}^{t+1}\left(x^{t+1},y^{t+1},b^{t+1}\right)}{\rho_{k}^{t}\left(x^{t+1},y^{t+1},b^{t+1}\right)}\right]}^{\frac{1}{2}}\\ \; & ={\left[\frac{\rho_{k}^{t+1}\left(x^{t},y^{t},b^{t}\right)}{\rho_{k}^{t}\left(x^{t},y^{t},b^{t}\right)}\times \frac{\rho_{k}^{t}\left(x^{t+1},y^{t},b^{t}\right)}{\rho_{k}^{t+1}\left(x^{t+1},y^{t},b^{t}\right)}\right]}^{\frac{1}{2}}\times {\left[\frac{\rho_{k}^{t}\left(x^{t+1},y^{t+1},b^{t+1}\right)}{\rho_{k}^{t+1}\left(x^{t+1},y^{t+1},b^{t+1}\right)}\times \frac{\rho_{k}^{t+1}\left(x^{t+1},y^{t},b^{t}\right)}{\rho_{k}^{t}\left(x^{t+1},y^{t},b^{t}\right)}\right]}^{\frac{1}{2}}\\ \; & =IBTC_{k}^{t,t+1}\times OBTC_{k}^{t,t+1} \end{array}$$i.e.,
(7)$$TC_{k}^{t,t+1}=MATC_{k}^{t,t+1}\times IBTC_{k}^{t,t+1}\times OBTC_{k}^{t,t+1}$$

*MATC* represents the scale effect of technological progress, refers to the neutral transfer of the technological frontier, while BTC means the bias of technological progress refers to the “non-neutral” transfer of the technological frontier. *IBTC* and *OBTC* reflect the impact of input and output changes on technological progress. If *IBTC* (*OBTC*) > 1 (<1), indicates progress (regression) in input-biased technology. When *IBTC* and *OBTC* = 1, it means that the technology change is Hicks-neutral.

It needs to be pointed out that output-based technological progress bias refers to the ability of technological progress to produce more output when the input factors are unchanged. We draw on the ideas of Weber and Domazlicky [50] and Li et al. [41] on the discriminative approach to the relationship between the direction of technological change and the elements. When $y_{1}^{t+1}/y_{2}^{t+1}>y_{1}^{t}/y_{2}^{t}$, *OBTC* > 1 indicates a $y_{2}$-producing biased technological change and *OBTC* < 1 indicates a $y_{1}$-producing biased technological change. When *OBTC* = 1, the output-biased technological change is Hicks-neutral. When $y_{1}^{t+1}/y_{2}^{t+1}<y_{1}^{t}/y_{2}^{t}$, *OBTC* > 1 indicates a $y_{1}$-producing biased technological change and *OBTC* < 1 indicates a $y_{2}$-producing biased technological change. The $y_{1}$-producing biased technological change means that output-biased technology tends to produce more $y_{1}$ relative to $y_{2}$, while the $y_{2}$-producing biased technological change tends to produce more $y_{2}$ relative to $y_{1}$.

This paper will focus on analyzing the output bias of technological progress. The specific descriptions of technical bias relationships are listed in Table 1. $y_{g}$ represents desired output; $y_{b}$ represents the undesired output.

### 2.2. The Regression Model

In general, the relationship between the impact of technological progress on environmental pollution has not yet reached a unified conclusion. One view is that the abatement effect of technological progress on pollutants cannot offset the environmental pollution caused by the increase in production efficiency brought about by technological progress. There is also a view that the reduction effect of technological progress and pollutants is in line with the inverted Kuznets U-curve for the environment. When a country’s technological progress is on the left side of the inflection point, and the technological progress itself is based on the fossil energy-based industrial system, the essence of its technological innovation is with pollution attributes. Its pollution reduction effect is necessarily limited. When the level of technological progress is on the right side of the inflection point, then the technological progress is biased toward the use of clean energy, with environmental attributes. The progress of technology can promote pollution reduction. According to the above idea, to further explore the relationship between technological progress and pollutants, the scatter fit of the relationship between technological progress and pollutants required for the analysis of this paper is shown in Figure 1. From Figure 1, an obvious non-linear relationship is seen between technological progress and CO_2_, SO_2_, and PM_2.5_. Therefore, the quadratic term of technological progress is introduced when constructing the mechanism of the effect of technological progress on pollutants to draw relevant conclusions in favor of reducing environmental pollution.

The model is set up as follows.
(8)$$LnPollution_{it}=\alpha_{0}+\alpha_{1}LnTC_{it}+\alpha_{2}{\left(LnTC\right)}^{2}+\alpha_{3}X_{it}+\varepsilon_{it}$$
where $Pollution_{it}$ are multiple pollutants such as CO_2_, SO_2_, and PM_2.5_, $TC_{it}$ is the technological progress, $X_{it}$ is the set of control variables for the t period of the i province, and $\varepsilon_{it}$ is the random error term of the econometric model. To avoid possible heteroskedasticity and the effect of differences in different variables, the model is logarithmized.

### 2.3. Data Sources

In this paper, we use Chinese provincial panel data for green total factor productivity measurement. Based on the availability and reliability of the data, we set the time span of the study as 2004–2018 and apply the DEA-SBM model to measure China’s green total factor productivity based on input-output data of 30 Chinese provinces (except Tibet), and decompose the technological progress and biased technological progress from green total factor productivity. The input-output data and processing are as follows.(1)Labor input. Labor input is measured using the number of employed persons in each province of China, and the data are obtained from the statistical yearbooks of each province of China.(2)Capital input. The amount of completed fixed asset investment in each province of China is used to measure capital input. The data are obtained from the statistical yearbooks of each province of China.(3)Energy input. We use the data of energy consumption of each province in China; the data come from the China Energy Statistical Yearbook.(4)Desired output. Nominal GDP data of each province in China are used and deflated to real GDP using CPI index, data from China Statistical Yearbook.(5)Undesired output. We use pollution data for each province in China, where CO_2_ is the converted emissions from energy consumption in each province, and the conversion method uses the carbon emission calculation method in the Guidelines for National Greenhouse Gas Inventories prepared by the Intergovernmental Panel on Climate Change (IPCC, 2016).

The calculation formula is:
$${\mathrm{CO}}_{2}=\sum_{i=1}^{8}{\mathrm{CO}}_{2,it}=\sum_{i=1}^{8}E_{it}\times NCV_{i}\times CEF_{i}$$
where CO_2_ represents the amount of carbon dioxide emissions to be estimated, i represents different types of energy, $E_{it}$ represents the combustion consumption of various energy sources, and $NCV_{i}$ represents the average low calorific value of various energy sources. The value comes from the China Energy Statistical Yearbook. $CEF_{i}$ represents the carbon dioxide emission factor of various energy sources, and the value comes from IPCC (2016).

SO_2_ is obtained from the industrial emission data of each province, and the data are obtained from the China Environmental Statistical Yearbook. PM_2.5_ is satellite gridded data measured by the Atmospheric Composition Analysis Group at Dalhousie University, Canada (https://fizz.phys.dal.ca/~atmos/martin/?page_id=140 (accessed on 22 July 2021)).

To determine the relationship between technological progress and pollutants, this paper also includes variables such as Economic Development Level, Energy Structure, Industrial Structure, Human Resources, and Foreign Direct Investment in the econometric model. 

Energy Structure (ES): The proportion of coal energy consumption to total energy consumption is used to measure the regional energy consumption structure. The higher the level of this variable, the less green the region is.

Industrial Structure (IS): Measured by the proportion of the output value of the secondary industry in the region’s GDP, the higher the level of this variable, the more the region’s economic development relies on the industry.

Human Capital (HC): Human capital refers to the degree of knowledge, skills, and quantity of labor in the workforce. Regions with higher human capital tend to have more advanced technological standards and higher environmental awareness. Higher human capital is beneficial to local environmental improvement, and this paper uses the number of years of education per employed person to measure the level of human capital.

Foreign Direct Investment (FDI): The current research findings on FDI are not uniform, and some studies suggest that foreign investment will cause the transfer of polluting industries from developed countries to developing countries, leading to environmental pollution in host countries [51,52]. Some studies also argue that foreign investment will promote the technological progress of host country enterprises and generate learning effects, which will reduce environmental pollution [53]. Therefore, the impact of foreign investment on the host country depends on factors such as whether the host country introduces polluting industries and its own technological level, and this paper uses the ratio of FDI amount to local GDP to measure the FDI level.

Economic Level (EL): There is a strong correlation between the level of economic development and the environmental pollution problem. Early economic development often came at the expense of the environment, which also led to serious environmental pollution in China, and with the economic model change in recent years, China’s environmental problems have not been properly solved. In this paper, the GDP per capita of each province in China is selected to measure the economic development level of the region.

The data of the above economic data variables are taken from the China Statistical Yearbook and the statistical yearbooks of each province, and the energy variables are taken from the China Environmental Statistical Yearbook, and the descriptive statistics of each indicator are shown in Table 2

## 3. Empirical Results

### 3.1. Green Total Factor Productivity and Its Decomposition Results

To better understand the technological progress of each province in China from 2004 to 2018, this paper measures the green total factor productivity of each province in China and decomposes it into technological progress and technical efficiency. Further, this paper decomposes technological progress into biased technological progress (BTC), including input-biased technological progress (IBTC) and output-biased technological progress (OBTC). The green total factor productivity and its decomposition terms for China as a whole and for each province are shown in Table 3, and the year-by-year averages are shown in Table 4 and Figure 2. In order to better understand the trend characteristics of green TFP and technological progress and technological progress bias in China, these indicators are cumulatively multiplied separately, and the processing is shown in Figure 3.

From Figure 2 and Figure 3, it can be seen that China’s overall green total factor productivity grew year by year during the study period, and its growth rate gradually increased from 2013 and turned around in 2017. During the period of 2013–2017, China’s economy developed rapidly, and foreign trade progressed rapidly, and the total imports and export increased from CNY 25,816.8 billion in 2013 to CNY 27,809,924 in 2017. The export competition and export learning effect during this period made China’s technological progress soar. Both in terms of trend and growth rate, it can be seen that the trend of TFP is roughly the same as that of TC, proving that technological progress is the main reason for the increase in total factor productivity. From the trend graph of biased technological progress (BTC) in Figure 3b, it can be seen that the degree of biased technological progress in China increases year by year and the output-based technological progress bias dominates compared to the input-based technological progress bias, which shows that BTC in China mainly focuses on the output side, aiming to put in fixed factors of production to produce as much output as possible. Therefore, this paper analyzes the output-biased technological progress in China and explores the directional changes in the output side in China.

### 3.2. Analysis of the Characteristics of Output Biased Technological Progress Factors

With the different needs of the times, China’s economic development model is also changing, which has different requirements for the technological progress behaviors such as technological innovation and technology introduction in the production process, i.e., the production process has different factor bias for the input process and output factor process. During the pursuit of high economic growth, this brash development model determined that the production process tended to ignore environmental protection, resulting in serious environmental pollution problems. In recent years, China’s economic development model has changed from pursuing high growth to the pursuit of sustainable and high-quality economic growth, and the demand for environmental protection has been gradually increasing. Therefore, an analysis of the elements of China’s technological progress bias will allow us to analyze whether China’s technological progress is free from “carbon lock-in” and whether there is a bias for pollutant reduction among pollutants. In order to better understand whether the bias of China’s technological progress has a bias among various pollutants, the average values of pollutant concentrations in each province of China were processed, as shown in Figure 4. From Figure 4, it can be seen that CO_2_ is still on a rising trend year by year, SO_2_ is on a decreasing trend in general, PM_2.5_ is on a rising trend first and then on a decreasing trend, and the inflection point of SO_2_ is later than the inflection point of PM_2.5_. These three pollutants differ in their reduction effects. They have heterogeneity in their reduction times, indicating a certain bias in reducing different pollutants at different stages of China’s economic development process. Based on the previous principle of output-based technological progress bias, we measured the characteristics of output technology bias elements for China as a whole and each province, as shown in Table 5. As seen in Table 5, the output of the production side of Chinese provinces is seen to fluctuate significantly during the study period, and CO_2_ output has been on an upward trend, while in contrast, SO_2_ and PM_2.5_ are clearly on a downward trend, with PM_2.5_ showing an upward and then a downward trend, a result that is corroborated by the changes in pollutant concentrations in China.

Due to the vast territory of China and the large differences in economic structure and technology levels among various regions, this paper summarizes the pollution emission bias of four major economic regions in China, including the northeast, central, eastern, and western regions, after dividing China into four major economic sub-regions, and finds that in general, the northeast, central, and western regions show GDP < CO_2_ < PM_2.5_ < SO_2_. The eastern region shows the output preference of PM_2.5_ < CO_2_ < SO_2_ < GDP. However, as time progresses, the output of PM_2.5_ and SO_2_ show a decreasing trend. In contrast, CO_2_ still shows an increasing trend, indicating that fossil combustion is still the main energy supply system in these regions. There is a “carbon lock-in effect” in technological progress. Still, the emission of pollutants such as PM_2.5_ and SO_2_ has decreased due to the due to China’s increased control of industrial emissions and the upgrading of environmental processes and equipment such as flue gas desulfurization treatment. There is also a shift in the pollution chain within China, with heavy polluting industries moving from economically developed regions to less economically developed regions, also contributing to pollution problems in inland provinces. The northeast, central, and western regions of China have always been economically less developed regions that rely mainly on industry, and the technology level and energy structure of these regions are less developed, and their pollution reduction capacity lags behind that of the eastern regions. The results of the above analyses show that although the overall emissions of pollutants such as PM_2.5_ and SO_2_ have been reduced in China, its production side still favors a production model with pollution attributes, and the reduction of its pollutant concentration is probably due to the increased control of pollutant production and the level of pollutant treatment processes, and CO_2_ emissions do not show a decreasing trend. This indicates that China’s overall energy structure is still dominated by fossil fuels as the main production system, and there is a certain “carbon lock-in effect” in its technological progress, which promotes the transformation of technology to cleaner production and energy conservation and environmental protection, while there may also be a certain pollution phenomenon caused by the expansion of production due to technological improvement.

### 3.3. Analysis of the Influencing Factors of Pollutants

Table 6 shows the test results of technological progress on heterogeneous pollutant emissions. The results show that CO_2_, PM_2.5_, SO_2_, and technological progress all show an inverted U-shaped relationship, indicating a significant increase and then decrease between pollutants and technological progress. The primary and secondary term coefficients of the variables all pass the significance level test. From the results of the model, the inflection point between technological progress and CO_2_ is 1.343, and the average value of LnTC in China’s provinces in 2018 is 1.130, which is still on the left side of the inflection point, so the current technological progress has not yet exerted an emission reduction effect on CO_2_. The inverted U-shaped inflection point of technological progress for SO_2_ is 1.08, and the current level of technological progress in China is already on the right side of the inflection point. The increase in the level of technological progress contributes to the reduction of SO_2_ concentration, which is also verified in the previous graph about SO_2_ concentration. The inverted U-shaped inflection point of technological progress for PM_2.5_ is 1.052, indicating that China’s technological level is on the right side of the inflection point. The improvement of technological progress can effectively reduce PM_2.5_ emissions. From the model results, there is a selective bias in the pollution emissions of technological progress in China. For CO_2_, technological advancement cannot reduce CO_2_ production, which indicates that there is a certain “carbon lock-in effect” in China’s current economic situation. However, technological advancement can effectively reduce SO_2_ and PM_2.5_ concentrations, probably because China started to control SO_2_ emissions in the early 20th century and has formulated a series of measures and policies to reduce SO_2_ pollution, and has technically realized the treatment of SO_2_ emissions and PM_2.5_ as the main pollutants of urban haze pollution, and its harm is self-evident. The government at all levels in China has made a lot of efforts to reduce haze pollution and achieve the goal of PM_2.5_ emission reduction. In terms of the influencing factors of pollutants, energy structure has a positive relationship with pollutant concentration, indicating that the larger the proportion of coal energy in energy consumption, the more serious the pollution. Industrial structure has a positive relationship with pollutant concentration, indicating that the greater the proportion of the industrial economy, the more serious the pollution. The coefficient of FDI is negative for CO_2_ and SO_2_ and positive for PM_2.5_, indicating that current foreign direct investment in China can reduce CO_2_ and SO_2_ pollution but increase PM_2.5_ pollution, which may be related to the production projects or production processes of foreign investment in China. The positive relationship between the level of economic development and CO_2_ and PM_2.5_ indicates that the phenomenon of economic development accompanied by environmental pollution still exists.

## 4. Discussions and Conclusions

Technological progress is of great importance in the process of pollution reduction. However, there are “carbon lock-in” and “energy rebound” effects in the relationship between technological progress and pollutants, increasing technological level not necessarily reducing the concentration of pollutants. To investigate the relationship between technological progress and pollutants in China, this paper uses the DEA-SBM model to measure the green total factor productivity of 30 Chinese provinces, decomposes the technological progress index from the green total factor productivity, and decomposes the technological progress index into output-based technological progress bias and input-based technological progress bias. Combining the results of China’s pollutant concentration map and output-based technological progress bias factors, this paper finds that biased emissions of pollutants exist in all regions of China. This result is corroborated by the analysis of the influencing factors of pollutants. Combining these analyses, the results show that:(1)There is a selection bias for pollutant emissions in China’s technological progress. CO_2_, PM_2.5_, and SO_2_ have an inverted U-shaped relationship with technological progress. As the level of technological progress increases, SO_2_ and PM_2.5_ emissions are significantly suppressed, and the level of technology breaks the inflection point of the inverted U-shape. As for CO_2_, the improvement in technology level has not yet played a significant role in reducing CO_2_ emissions, which indicates that there is a certain “carbon locking effect” and “energy rebound effect” in China’s technological progress, and also indicates that China has not yet shifted from its crude economic development model.(2)From the perspective of the output-based technological progress bias factor, it can be seen that the output-based technological progress bias in CO_2_ output is steadily increasing, SO_2_ is decreasing, and PM_2.5_ is increasing and then decreasing. This is corroborated by the changes in pollutant concentrations. However, due to the vast geographical area of China, there are obvious differences in economic structure and technological level among different regions, and their output-based technological progress bias factors also differ. In general, the northeast, central, and western regions show the output bias of GDP < CO_2_ < PM_2.5_ < SO_2_, while the eastern region shows the output preference of PM_2.5_ < CO_2_ < SO_2_ < GDP. However, as time progresses, the output of both PM_2.5_ and SO_2_ shows a decreasing trend, while CO_2_ still shows an increasing trend, indicating that fossil combustion is still the main energy supply system in these regions, and there is a “carbon lock-in effect” in technological progress, and China’s emission treatment technologies for both PM_2.5_ and SO_2_ are more mature, achieving the emission of both pollutants. China’s emission treatment technology for PM_2.5_ and SO_2_ is more mature and has achieved the reduction of these two pollutants.(3)In terms of pollutant influencing factors, energy structure and industrial structure are positively related to pollutants, and its economic development is also positively related to CO_2_ and PM_2.5_, indicating that China’s technological progress has a carbon lock-in effect and economic development is still accompanied by environmental pollution, indicating that China has not ended the brutal economic development mode during the study period and is still in the process of economic transformation.

This paper analyzes green total factor productivity and technological progress in China, and investigates the output bias of technological progress on the output relationship of multiple pollutants. The findings effectively enrich the research related to technological progress and environmental protection, and provide theoretical support for the Chinese government to adjust its environmental protection and production policies. Based on the results of the above analysis, this paper gives the following recommendations.(1)Strengthen government support to guide technological innovation. The government should fully support technological innovation and other behaviors in terms of policies, such as subsidies and tax incentives for R&D and application of clean technologies, to promote technological progress in each region to cross the inverted U-shaped inflection point in carbon emissions. At the same time, the government should actively guide research institutes and other research institutes to cooperate with enterprises to better transform and apply clean technologies.(2)Increase the proportion of new energy applications and eliminate the carbon lock-in effect. China’s economic model is still in transition from the pursuit of economic growth to high-quality development, and the pollution problems accompanying economic development have existed for a long time, with CO_2_ emissions being the primary target of all pollutants. The reduction of SO_2_ and PM_2.5_ pollution concentrations is attributed to the Chinese government controlling these two pollutants’ emissions and the technological progress of pollutant treatment processes. However, there are still pollutant outputs at the source of production. Moreover, from the results of this paper, there is a “carbon lock-in effect” in China’s economy. The coal-based energy consumption structure makes the emission of CO_2_ and other pollutants more serious, and increasing the proportion of new energy applications such as photovoltaic, wind power, and hydropower can effectively reduce the output of pollutants on the production side.(3)Promote the development of technology to be clean and green. China’s existing production system is based on fossil energy. Transitioning to clean, green development can effectively reduce the pollutant emissions in the production process, as can strengthening technology research and development, enhancing the learning effect of foreign investment, and taking other actions to help improve the level of technological progress.(4)Actively guide the technology exchange between regions and promote the transformation of industries to a low-pollution direction. The northeast, central, and western regions show an overall preference for pollutant emissions, which is due to the less developed production levels dominating these regions. Actively guiding these regions and the developed eastern regions in technology exchange and cooperation, breaking the technical barriers between regions, and promoting the transfer and diffusion of advanced clean production technologies from the eastern regions to these regions will help these regions to improve their technology levels. The government should promote the transformation of industries in a low-pollution direction and promote the transformation of regional industries in a green direction, facing the obvious differences in economic structure and technology level between the regions that exist. To promote the transformation of regions with underdeveloped production capacity and polluting industries to low-pollution production methods, the government should introduce policies including subsidies for the renewal of clean equipment so that regions with polluting industries can end the status of rough economic development.

## Figures and Tables

**Figure 1 ijerph-18-09286-f001:**
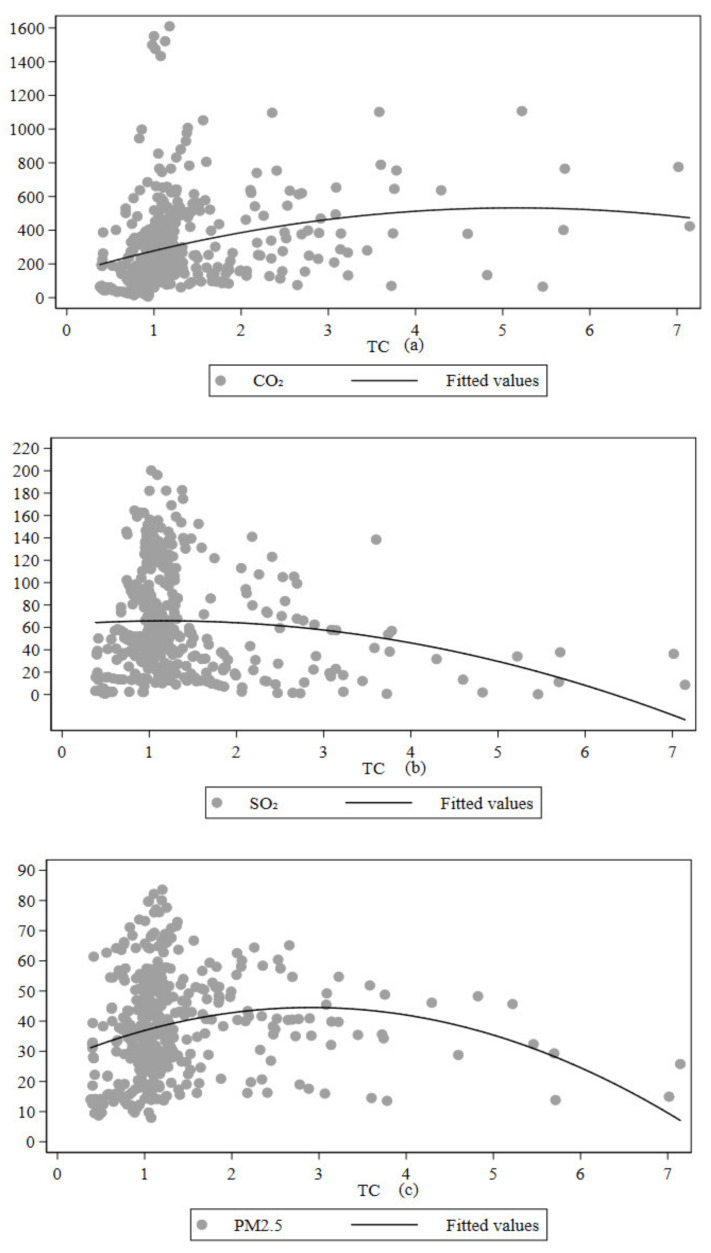
Scatterplot of technological progress and pollutants: (**a**) CO_2_ and technological progress, (**b**) SO_2_ and technological progress, (**c**) PM_2.5_ and technological progress.

**Figure 2 ijerph-18-09286-f002:**
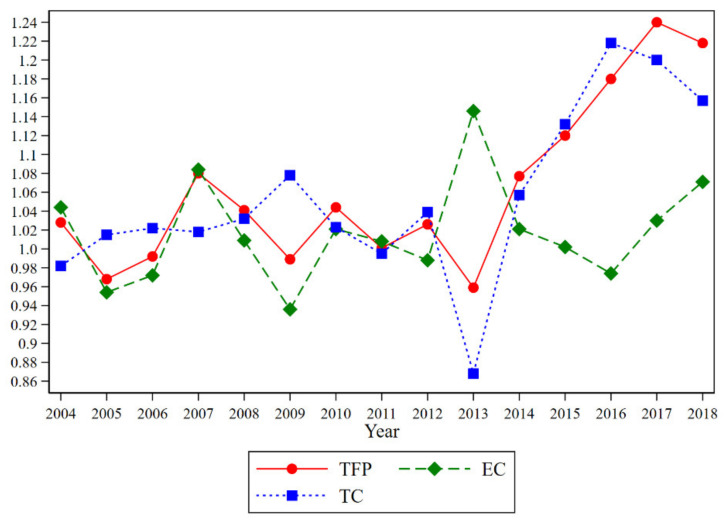
China’s green total factor productivity and its decomposition term growth rate.

**Figure 3 ijerph-18-09286-f003:**
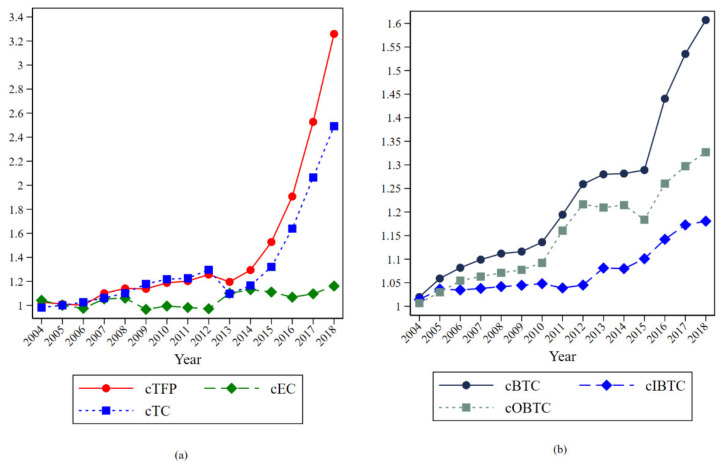
Trend of cumulative green total factor productivity and its decomposition term in China: (**a**) trend in the cumulative index of TFP, EC and TC. (**b**) trend in the cumulative index of BTC IBTC and OBTC.

**Figure 4 ijerph-18-09286-f004:**
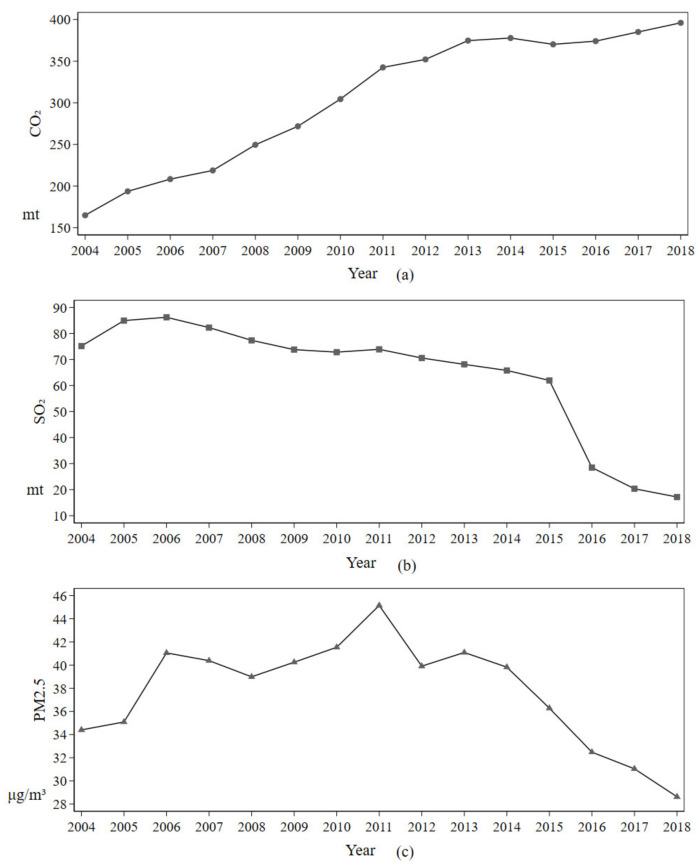
Pollutant mean value graph: (**a**) CO_2_ yearly average. (**b**) SO_2_ yearly average. (**c**) PM_2.5_ yearly average.

**Table 1 ijerph-18-09286-t001:** Biased technical change direction in output mix.

**Output Mix**	OBTC > 1	OBTC = 1	OBTC < 1
ybt+1ygt+1>ybtygt	Promote desirable output	Neutral	Increase undesirable output
ybt+1ygt+1<ybtygt	Increase undesirable output	Neutral	Promote desirable output

**Table 2 ijerph-18-09286-t002:** Descriptive statistical analysis of variables.

	Variable	Description of Variable	Unit	Average	Max	Min	SD
Inputs and outputs	L	Labor	Ten thousand-person	490.136	1994.140	42.520	337.815
	K	Capital	100 million yuan	32,478.120	183,575.000	962.342	31,507.960
	E	Energy Consumption	Ten thousand tons	12,764.150	40,581.000	742.000	8231.907
	GDP	Gross Domestic Product	100 million yuan	12,464.540	69,174.320	430.179	11,508.100
	CO_2_	Carbon Dioxide Emission	Ten thousand tons	305.615	1609.710	7.550	253.756
	SO_2_	Sulfur Dioxide Emission	Ten thousand tons	63.933	200.300	0.267	45.028
	PM_2.5_	Particulate Matter	Micrograms per cubic meter	37.741	83.608	7.953	16.329
Influencing Factors	ES	Energy Structure		0.4398043	0.440	0.778	0.016
	IS	Industrial Structure		44.10384	44.104	62.000	16.500
	HC	Human Capital		9.362649	9.363	13.617	6.429
	FDI	Foreign Direct Investment		0.0250811	0.025	0.121	0.000
	EL	Economic Level		28,119.75	28,119.750	110,580.300	4081.554

**Table 3 ijerph-18-09286-t003:** Descriptive statistics of total factor productivity and its decomposition term for Chinese provinces.

Area	TFP		EC		TC		OBTC		IBTC		MATC		BTC	
	Value	S.D	Value	S.D	Value	S.D	Value	S.D	Value	S.D	Value	S.D	Value	S.D
China	1.032	0.044	1.000	0.000	1.032	0.044	0.998	0.056	1.022	0.058	1.016	0.073	1.018	0.033
Beijing	1.130	0.165	1.000	0.000	1.130	0.165	1.042	0.092	1.010	0.046	1.074	0.116	1.055	0.123
Tianjin	1.149	0.354	1.039	0.329	1.127	0.218	1.094	0.192	1.003	0.004	1.026	0.040	1.096	0.194
Hebei	1.093	0.286	1.003	0.132	1.082	0.147	0.998	0.035	1.036	0.138	1.049	0.092	1.030	0.090
Shanxi	1.015	0.082	1.003	0.074	1.012	0.034	1.002	0.003	1.001	0.001	1.009	0.033	1.003	0.003
Inner Mongolia	1.257	0.323	1.170	0.583	1.213	0.391	1.019	0.271	0.974	0.055	1.316	0.613	0.994	0.283
Liaoning	1.032	0.071	0.997	0.053	1.035	0.061	1.003	0.004	1.001	0.001	1.031	0.060	1.004	0.004
Jilin	1.020	0.062	1.004	0.078	1.019	0.053	0.999	0.016	1.002	0.001	1.018	0.053	1.002	0.016
Heilongjiang	1.045	0.350	1.032	0.320	1.020	0.152	0.974	0.099	1.012	0.023	1.049	0.217	0.985	0.103
Shanghai	1.071	0.071	1.000	0.000	1.071	0.071	0.954	0.056	1.019	0.045	1.111	0.137	0.971	0.054
Jiangsu	1.135	0.141	1.026	0.101	1.109	0.120	0.919	0.045	0.992	0.036	1.226	0.186	0.911	0.061
Zhejiang	1.147	0.139	1.008	0.132	1.148	0.135	1.003	0.103	0.997	0.037	1.159	0.173	0.998	0.088
Anhui	1.092	0.258	1.025	0.288	1.126	0.338	0.973	0.135	1.039	0.146	1.161	0.427	1.001	0.130
Fujian	1.078	0.129	1.007	0.134	1.076	0.114	0.953	0.062	0.994	0.030	1.148	0.204	0.948	0.076
Jiangxi	1.037	0.045	1.000	0.040	1.038	0.038	1.004	0.006	1.002	0.005	1.032	0.035	1.006	0.004
Shandong	1.159	0.263	1.037	0.234	1.152	0.298	1.015	0.087	1.049	0.120	1.098	0.304	1.061	0.124
Henan	0.992	0.154	0.957	0.120	1.033	0.058	1.001	0.014	1.006	0.016	1.026	0.059	1.007	0.020
Hubei	1.164	0.270	1.066	0.225	1.095	0.161	1.033	0.105	1.011	0.032	1.047	0.063	1.043	0.095
Hunan	1.138	0.224	1.049	0.182	1.088	0.144	0.948	0.087	1.001	0.029	1.177	0.311	0.950	0.104
Guangdong	1.029	0.031	1.000	0.000	1.029	0.031	0.989	0.039	1.037	0.048	1.006	0.052	1.024	0.031
Guangxi	0.966	0.143	0.951	0.100	1.012	0.079	1.004	0.016	1.020	0.072	0.995	0.111	1.024	0.072
Hainan	0.960	0.113	1.000	0.000	0.960	0.113	1.135	0.096	1.023	0.030	0.835	0.120	1.159	0.091
Chungking	1.123	0.224	1.052	0.177	1.068	0.128	1.015	0.110	1.023	0.077	1.031	0.047	1.033	0.083
Sichuan	1.150	0.208	1.060	0.143	1.087	0.148	1.050	0.099	1.007	0.023	1.027	0.086	1.057	0.099
Guizhou	1.018	0.039	1.015	0.037	1.003	0.039	1.002	0.004	1.002	0.001	0.999	0.036	1.004	0.004
Yunnan	1.000	0.212	0.997	0.244	1.025	0.144	0.976	0.063	1.027	0.069	1.033	0.194	0.999	0.048
Shaanxi	0.967	0.134	0.970	0.113	0.998	0.079	1.034	0.097	0.978	0.068	0.991	0.050	1.006	0.048
Gansu	1.007	0.037	0.995	0.041	1.013	0.037	1.003	0.003	1.002	0.001	1.008	0.036	1.004	0.003
Qinghai	0.952	0.079	1.000	0.000	0.952	0.079	1.107	0.061	1.017	0.037	0.846	0.056	1.124	0.055
Ningxia	1.009	0.202	1.064	0.179	0.946	0.070	0.972	0.099	1.032	0.115	0.951	0.034	0.994	0.058
Sinkiang	0.997	0.068	0.993	0.060	1.004	0.033	1.003	0.005	1.001	0.001	0.999	0.032	1.005	0.005
Total	1.063	0.194	1.017	0.181	1.055	0.157	1.007	0.099	1.011	0.060	1.048	0.202	1.017	0.102

**Table 4 ijerph-18-09286-t004:** Annual average of green total factor productivity and its decomposition term in China.

Year	TFP	EC	TC	OBTC	IBTC	MATC	BTC
2004	1.028	1.044	0.982	1.005	1.016	0.970	1.020
2005	0.968	0.954	1.015	1.018	1.020	0.990	1.035
2006	0.992	0.972	1.022	1.018	1.000	1.009	1.018
2007	1.080	1.084	1.018	1.005	1.003	1.019	1.009
2008	1.041	1.009	1.032	1.001	1.005	1.028	1.006
2009	0.989	0.936	1.078	0.996	1.001	1.092	0.996
2010	1.044	1.021	1.023	1.005	1.002	1.019	1.006
2011	1.001	1.008	0.995	1.044	0.992	0.971	1.034
2012	1.026	0.988	1.039	1.029	1.005	1.004	1.034
2013	0.959	1.146	0.868	0.983	1.026	0.869	1.006
2014	1.077	1.021	1.057	0.990	1.001	1.076	0.990
2015	1.120	1.002	1.132	0.973	1.020	1.168	0.993
2016	1.180	0.974	1.218	1.034	1.033	1.173	1.067
2017	1.240	1.030	1.200	1.003	1.028	1.192	1.029
2018	1.218	1.071	1.157	1.006	1.006	1.158	1.007
Total	1.064	1.017	1.056	1.007	1.011	1.049	1.017

**Table 5 ijerph-18-09286-t005:** Output factor bias results.

Year	CO_2_ vs. GDP		CO_2_ vs. SO_2_		CO_2_ vs. PM_2.5_		SO_2_ vs. PM_2.5_	
	CO_2_	GDP	CO_2_	SO_2_	CO_2_	PM_2.5_	SO_2_	PM_2.5_
2004	9	20	14	15	12	17	12	17
2005	12	9	10	11	10	11	8	13
2006	21	9	6	24	16	14	20	10
2007	22	8	10	20	15	15	18	12
2008	15	15	7	23	10	20	16	14
2009	18	11	10	19	8	21	15	14
2010	18	10	7	21	10	18	18	10
2011	15	13	7	21	13	15	17	11
2012	18	9	7	20	8	19	10	17
2013	13	15	18	10	23	5	20	8
2014	15	13	13	15	11	17	12	16
2015	11	13	12	12	8	16	5	19
2016	15	15	10	20	11	19	20	10
2017	20	9	13	16	13	16	15	14
2018	19	11	12	18	17	13	16	14

**Table 6 ijerph-18-09286-t006:** Regression results of influencing factors.

	LnCO_2_	LnSO_2_	LnPM_2.5_
InTC	1.774 ***	4.047 ***	0.981 ***
	(−0.633)	(−0.652)	(−0.299)
(LnTC)^2^	−0.660 **	−1.873 ***	−0.466 ***
	(−0.289)	(−0.298)	(−0.137)
LnES	3.429 ***	3.731 ***	1.829 ***
	(−0.514)	(−0.53)	(−0.243)
LnIS	1.501 ***	1.848 ***	0.281 ***
	(−0.218)	(−0.224)	(−0.103)
LnHC	1.045	−0.935	0.504
	(−0.649)	(−0.669)	(−0.307)
LnFDI	−11.78 ***	−5.114 **	5.331 ***
	(−2.057)	(−2.119)	(−0.971)
LnEL	0.678 ***	0.0918	0.222 ***
	(−0.15)	(−0.154)	(−0.0706)
_cons	−11.30 ***	−4.917 ***	−2.147 ***
	(−1.285)	(−1.324)	(−0.607)

Note: **, and *** represent that the estimated coefficient is significant at 5%, and 1% confidence levels, respectively. The standard errors of the coefficients are marked in parentheses.

## Data Availability

The raw/processed data required to reproduce these findings cannotbe shared at this time as the data also forms part of an ongoing study.

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
