# Peer review of "Is Technological Progress Selective for Multiple Pollutant Emissions?"

_ijerph, 2021, doi:10.3390/ijerph18179286_

Round 1
Reviewer 1 Report
The authors submitted a manuscript entitled “Are technological progress selective for multiple pollutant emissions? --An inverted U-shaped test for multiple pollutants in China” to International JERPH Journal of Environmental Research and Public Health with the reference ijerph-1339325.
The subject is a very interesting one, important nowadays, the methodology is well explained (no new methods), the results are clearly presented and appear solid and a huge effort was made to extract important meaning from the results.
The manuscript could be accepted but some minor changes should be done.
The English language is understandable but many times close by phrases do repeat the same idea. As an example please see lines 384-386. Trying to avoid this will make the text lighter and more readable.
The Abstract is too long. An effort to shorten it should be done. In addition, on lines 23-24 part of a phrase is repeated. This must be corrected.
Some conclusions are built based on the form of the inverted U-shape curve shown I Figure 1. Looking to this figure is clear that the point are quite scattered and that other fits are also possible. Probably it is out of scope to use other models in this paper, but at least some reference to these uncertainties due to the scatter of points should be introduced in the Discussion and Conclusions. Otherwise, the readers could assume there is no doubt about conclusions (e.g. in which leg, ascending or descending the pollutants are) and that would not correspond to the true.
Colors in Figures are clear and appropriate. Tables are quite readable.
Reviewer 2 Report
After reading the manuscript “Are technological progress selective for multiple pollutant emissions? --An inverted U-shaped test for multiple pollutants in China”, I highlight next comments:
- The length of Abstract cannot exceed a maximum of 200 words.
- Background is weak, most arguments are based on environmental pollution but the role of technological progress in pollutant emissions was barely examined despite the statement of lines 198-199. Literature review disregarded relevant global studies in the matter. Scope of the research was extensively described in lines 99 to 119, but research aim was not clearly determined. Rather than lines 112 to 119, actual contributions in the field were not disclosed either.
- Section 2 mostly involves the DEA-SBM and the regression models, but the relationship with the U-shaped test was ignored. Consequently, the title of the article seems misleading. Rationale to select those models is vague. A general overview that explains all methodological stages by linking them to the models presented is lacking. A chart is thus recommended. Are all mathematical expressions necessary? I don´t believe so. Only critical equations required to enable the replication of the study should be given. Criteria used to establish Table 1 as the basis of the study are unknown. Same to the model expressed in equation 8 and chats of Figure 1. Why does the research examine the period from 2004 to 2018? A set of variables was described in subsection 2.3 (Table 2), but its correlation with the two models previously proposed is unclear.
- The breakdown of the green total factor productivity of each Chinese province was exhibited in Table 3, but the expression of the composite factor was not provided. Indeed, the application of models given in Section 2 to obtain values of Tables/Figures of Section 3 is very unlikely
- Beyond the statistical analysis displayed in Section 3, discussion must also connect results in relation to other analyses in the same line conducted in other countries and the factors associated to the technological process of the 30 provinces studied in the last decades. Responses to questions posed in 83 to 89 were not provided.
- Recommendations seem arbitrary without a clear linkage with findings. Conclusions that outline theoretical/practical implications in the field were omitted.
- Miscellaneous comments. Acronyms must be preceded by the full description at first appearance. Some references seem incomplete such as [3] and [6]. English grammar and style should be improved.
Several mathematical expressions were given with scarce theoretical background and correlation to be applied to data from 30 Chinese provinces in the period from 2004 to 2018. But methodological shortcomings hinder the replicability of the proposed models. Implications in the field are ambiguous. Authors are invited to enhanced theoretical and methodological frames.
Round 2
Reviewer 2 Report
After reading the revised manuscript “Is technological progress selective for multiple pollutant emissions? --An inverted U-shaped test for multiple pollutants in China”, I highlight next comments:
- The length of Abstract cannot exceed a maximum of 200 words.
- Literature review should be enhanced to strengthen the weak background. Research aim remains unspecified alongside theoretical/practical contributions in the field.
- A general overview that explains all methodological stages by linking them to the models presented is lacking. Relationship between U-shaped test and DEA-SBM & regression models was not disclosed. Only critical equations required to enable the replication of the study should be given. Criteria used to establish Table 1 as the basis of the study should be displayed. Same to the model expressed in equation 8 and charts of Figure 1. A set of variables was described in subsection 2.3 (Table 2), but its correlation with the two models previously proposed is unclear.
- The breakdown of the green total factor productivity of each Chinese province was exhibited in Table 3, but the expression of the composite factor was not provided. Indeed, the application of models given in Section 2 to obtain values of Tables/Figures of Section 3 is very unlikely
- Beyond the statistical analyses displayed in Section 3, discussion must also connect results in relation to other analyses in the same line conducted in other countries and the factors associated to the technological process of the 30 provinces studied in the last decades. Responses to questions posed in the first section were not provided.
- Recommendations are not grounded on findings. Theoretical/practical implications in the field were omitted in the last section.
- Miscellaneous comments. Concise Point-to-point responses to reviewer´s remarks were not provided. Remarks must be addressed in the manuscript, the cover letter is not sufficient at all. Acronyms must be preceded by the full description at first appearance. Some references seem incomplete like [3].
